# `Quark`: Controllable Text Generation with Reinforced [Un]learning

**Ximing Lu**[♠♡]     **Sean Welleck**[♠♡∗]     **Jack Hessel**[♡∗]     **Liwei Jiang**[♠♡]
**Lianhui Qin**[♠]     **Peter West**[♠]     **Prithviraj Ammanabrolu**[♡]     **Yejin Choi**[♠♡]
[♡]Allen Institute for Artificial Intelligence
[♠]Paul G. Allen School of Computer Science, University of Washington
{ximinglu, jackh, raja}@allenai.org
{wellecks, lwjiang, lianhuiq, pawest, yejin}@cs.washington.edu

https://github.com/GXimingLu/Quark

## Abstract

Large-scale language models often learn behaviors that are misaligned with user expectations. Generated text may contain offensive or toxic language, contain significant repetition, or be of a different sentiment than desired by the user. We consider the task of *unlearning* these misalignments by fine-tuning the language model on signals of what *not* to do. We introduce `Quantized Reward Konditioning` (`Quark`), an algorithm for optimizing a reward function that quantifies an (un)wanted property, while not straying too far from the original model. `Quark` alternates between (i) collecting samples with the current language model, (ii) sorting them into quantiles based on reward, with each quantile identified by a reward token prepended to the language model's input, and (iii) using a standard language modeling loss on samples from each quantile conditioned on its reward token, while remaining nearby the original language model via a KL-divergence penalty. By conditioning on a high-reward token at generation time, the model generates text that exhibits less of the unwanted property. For unlearning toxicity, negative sentiment, and repetition, our experiments show that `Quark` outperforms both strong baselines and state-of-the-art reinforcement learning methods like PPO [66], while relying only on standard language modeling primitives.

## 1 Introduction

Large neural language models trained on an enormous amount of web text have excelled at numerous tasks [58, 87, 10]. They provide an effective interface for few-shot learning [8], show impressive natural-language understanding capabilities [47], and, in some contexts, their generations can be indistinguishable from human-authored text [11].

However, these same language models often exhibit undesirable behaviors, as they are usually trained to simply maximize the likelihood of their raw pre-training data. For example, models sometimes generate toxic text that reflects pernicious social biases [18, 69], or generate repetitive and dull language [79, 38, 25]. Undesirable behaviors are diverse and hard to avoid, control, or even specify *a priori*; we thus argue that it is critical to investigate ways to *unlearn* undesirable behaviors *post hoc*, while maintaining capacity for generating coherent and fluent language.

Supervised approaches for unlearning pose challenges. One option is to curate and train on a corpus that encodes desirable behavior, with the hope that additional maximum likelihood training will shape

---

[∗]equal contribution

36th Conference on Neural Information Processing Systems (NeurIPS 2022).

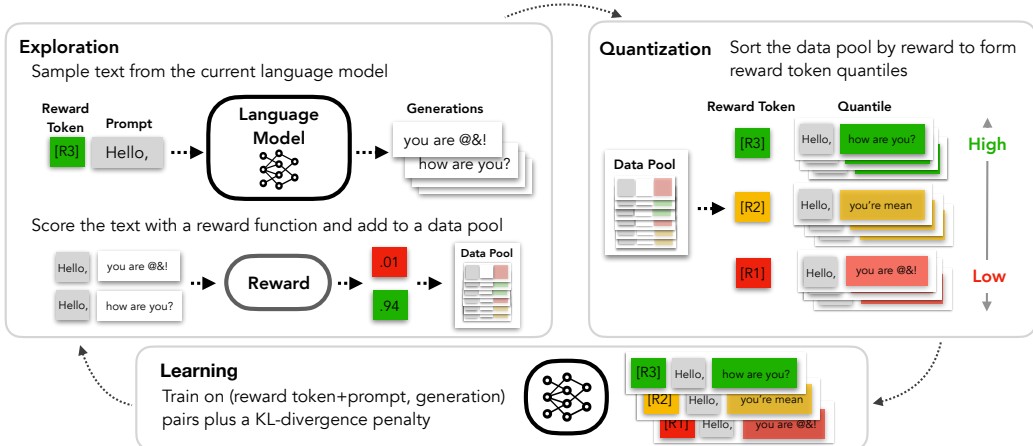

Figure 1: Quantized Reward Konditioning (Quark) is an online, off-policy reinforcement learning (RL) algorithm used to (un)learn properties from language models via three iterative stages: exploration, quantization, and learning.

the model's distribution more favorably. However, collecting data that accurately captures desired characteristics (e.g., non-toxic, non-degenerate texts) is difficult (if not impossible) [40]. Moreover, models may overfit to the newly collected corpora [40, 32] and lose desirable characteristics, e.g., few shot learning capacity over general domains. Another option is to build a detector of the undesirable behavior, e.g., by labelling model outputs. However, it is not clear how to adjust the model so that it only generates text that the detector prefers: since detectors score full text samples from the model rather than providing token-by-token feedback, they are not directly differentiable (e.g., toxicity scores) [54].

Dynamically (un)learning from sentence-level, scalar feedback is perhaps better suited to the reinforcement learning (RL) paradigm. In NLP, RL has been used to optimize scalar metrics in the form of rewards [54, 60, 83]. Recently [51] used Proximal Policy Optimization (PPO) [66] to optimize a 175B parameter model via a learned reward model, while constraining the model to remain close to the original with a KL-divergence penalty. However, as (deep) RL is highly sensitive to variance in the reward function [1, 41], these methods rely on additional models – often doubling the number of learnable parameters – and specialized heuristics to stabilize training.

We introduce Quantized Reward Konditioning (Quark), an algorithm for reward-based (un)learning with language models. Quark builds upon insights from three prior works: the Decision Transformer [9], LM tuning with PPO [91], and control tokens [28]. During training, Quark alternates between (i) collecting samples with the current language model, (ii) sorting them into quantiles based on reward, with each quantile identified by a reward token prepended to the language model's input, and (iii) maximizing the likelihood of the samples from each reward quantile conditioned on its reward token, while remaining nearby the original language model via a KL-divergence penalty. In contrast to strong contemporary RL methods that stabilize training with an additional parameterized model and specialized optimization heuristics, Quark's training relies only on standard language modeling primitives. Experiments across three tasks demonstrate that Quark maintains pre-training abilities while unlearning undesired behaviors more stably than alternative methods.

## 2   Quark: Quantized Reward Konditioning

Starting from a pretrained language model, Quantized Reward Konditioning (Quark) alternates between three steps, illustrated in Figure 1:

- **Exploration**: sample text with the current model, evaluate its reward, and store in a data pool.
- **Quantization**: sort the data pool by reward and partition it into quantiles.
- **Learning**: update the language model using samples from each quantile.

By sampling from high reward quantiles during exploration and using a KL-divergence penalty during learning, Quark iteratively improves the language model by steering its distribution towards

**Algorithm 1** Quantized Reward Konditioning (Quark)

---
**input** Initial policy $p_0$, prompts $X$, reward $r(\cdot)$, KL weight $\beta$, number of quantiles $K$

1: Make a copy $p_\theta$ of initial policy $p_0$; and `Initialize` data pool $\mathcal{D}$          ▷ Initialization
2: **for** iteration $= 1, 2, \ldots, N$ **do**
3:    **for** $x_i \in X$ **do**
4:      Sample generation $y_i \sim p_\theta(\cdot|x_i, r_K)$          ▷ Exploration
5:      Add $\big(x_i, y_i, r(x_i, y_i)\big)$ into data pool $\mathcal{D}$
6:    $\tilde{\mathcal{D}}_i \leftarrow$ `quantize`$(\mathcal{D}; K)$          ▷ Quantization
7:    **for** step $= 1, 2, \ldots, M$ **do**
8:      Draw a batch of data $\big\{(x_i, y_i, r_{ki})\big\}$ from quantized data pool $\tilde{\mathcal{D}}_i$          ▷ Learning
9:      Compute the objectives in Eq. 2
10:     Update the policy parameters $\theta$ via gradient descent

---

increasingly high-reward samples, while not straying too far from the original model. Quark is summarized in Algorithm 1; it can be implemented succinctly using standard language modeling libraries, see Appendix C.

**Initialization.** Quark begins with a pretrained language model $p_0(y|x)$, a set of training prompts $X$ and a reward function $r(x, y) \to \mathbb{R}$. Here $x = (x_1, \ldots, x_{|x|})$ and $y = (y_1, \ldots, y_{|y|})$ are sequences of tokens from a vocabulary $\mathcal{V}$. Quark initializes a *datapool* of (input, output, reward) examples by sampling[2] from $p_0$ conditioned on the training prompts, and scoring them with the reward function,

$$\mathcal{D}_0 = \{(x, y, r(x, y)) \mid y \sim p_0(\cdot|x), \text{ for all } x \in X)\}. \tag{1}$$

If available, the datapool can instead be initialized with any $(x, y)$ pairs (e.g., from a supervised dataset). Quark then proceeds iteratively, updating a copy of the pretrained language model, $p_\theta$, by alternating between *exploration*, *quantization* and *learning*. We detail quantization first.

**Quantization.** Quark quantizes each example in the datapool based on how high its reward is compared to others in the data pool. Quark sorts the current iteration's datapool in order of increasing reward, and partitions the sorted pool into equally sized quantiles, $\mathcal{D}^1, \ldots, \mathcal{D}^K$. Each sample $(x, y)$ is now part of a quantile that is identified by a reward token $r_k$ with $k \in \{1, \ldots, K\}$. For example, in Figure 1 the non-toxic generation *how are you?* is placed in the highest-reward quantile, identified by $r_3$, while the toxic generation, *you are \*@&!*, is placed in the lowest-reward quantile $r_1$.

**Learning.** For learning, Quark trains on the quantized datapool $\mathcal{D}$ using a standard conditional language modeling objective – maximizing likelihood – along with a KL-penalty to keep the model from deviating too far from the original:

$$\max_\theta \mathbb{E}_{k \sim \mathcal{U}(1, K)} \mathbb{E}_{(x, y) \sim \mathcal{D}^k} \left[ \log p_\theta(y|x, r_k) - \beta \sum_{t=1}^{T} \text{KL}\left(p_0(\cdot|y_{<t}, x) \| p_\theta(\cdot|y_{<t}, x, r_k)\right) \right], \tag{2}$$

where each KL term is $\sum_{y_t \in \mathcal{V}} p_0(y_t) \log \frac{p_0(y_t)}{p_\theta(y_t)}$ (omitting the conditioned terms). Naturally, Quark supports other penalties developed for language modeling, e.g., entropy [43] or unlikelihood [79].

**Exploration.** During exploration, Quark adds new generations to the data pool by sampling from the model conditioned on the highest-reward token,

$$\mathcal{D} \leftarrow \mathcal{D} \cup \{(x, y, r(x, y)) \mid y \sim p_\theta(\cdot|x, r_K), \text{ for all } x \in X\}, \tag{3}$$

where $y \sim p_\theta(\cdot|x, r_K)$ means sampling from the current model $p_\theta$, with the reward token $r_K$ prepended to the training input $x$. Intuitively, this step explores the most promising regions of the distribution by querying the current model for what it expects to be high reward completions.

**Evaluation.** At test time, we condition the language model on the highest reward token, $y \sim p_\theta(\cdot|x, r_K)$, and evaluate the resulting samples.

---

[2]Any decoding method can be used, e.g., greedy search, beam search, nucleus sampling [25].

| Model | In-domain (REALTOXICITYPROMPTS) | | | | | Out-of-domain (WRITINGPROMPTS) | | | | |
|---|---|---|---|---|---|---|---|---|---|---|
| | Toxicity (↓) | | Fluency (↓) | Diversity (↑) | | Toxicity (↓) | | Fluency (↓) | Diversity (↑) | |
| | avg. max. | prob. | output ppl | dist-2 | dist-3 | avg. max. | prob. | output ppl | dist-2 | dist-3 |
| GPT2 [57] | 0.527 | 0.520 | 11.31 | 0.85 | 0.85 | 0.572 | 0.610 | 12.99 | 0.82 | 0.85 |
| PPLM [12] | 0.520 | 0.518 | 32.58 | 0.86 | 0.86 | 0.544 | 0.590 | 36.20 | 0.87 | 0.86 |
| GeDi [32] | 0.363 | 0.217 | 60.03 | 0.84 | 0.83 | 0.261 | 0.050 | 91.16 | 0.86 | 0.82 |
| DEXPERTS [40] | 0.314 | 0.128 | 32.41 | 0.84 | 0.84 | 0.343 | 0.156 | 42.53 | 0.86 | 0.85 |
| DAPT [21] | 0.428 | 0.360 | 31.21 | 0.84 | 0.84 | 0.442 | 0.363 | 38.11 | 0.86 | 0.85 |
| PPO [71] | 0.218 | 0.044 | 14.27 | 0.80 | 0.84 | 0.234 | 0.048 | 15.49 | 0.81 | 0.84 |
| Quark | **0.196** | **0.035** | **12.47** | 0.80 | 0.84 | **0.193** | **0.018** | **14.49** | 0.82 | 0.85 |

Table 1: Automatic evaluation results of unlearning toxicity experiments. Baseline results (except PPO) are from [40].

| | Ours vs. GPT2 | | Ours vs. PPLM | | Ours vs. GeDi | | Ours vs. DEXPERT | | Ours vs. DAPT | | Ours vs. PPO | |
|---|---|---|---|---|---|---|---|---|---|---|---|---|
| | **In-domain** (REALTOXICITYPROMPTS) | | | | | | | | | | | |
| **Less Toxic** | **0.21** | 0.07 | **0.20** | 0.08 | **0.15** | 0.06 | **0.14** | 0.10 | 0.12 | 0.12 | 0.12 | 0.12 |
| **More Topical** | **0.22** | 0.14 | **0.23** | 0.14 | **0.21** | 0.13 | 0.18 | 0.18 | **0.20** | 0.16 | **0.22** | 0.14 |
| **More Fluent** | **0.26** | 0.19 | **0.27** | 0.17 | **0.29** | 0.15 | **0.26** | 0.21 | **0.23** | 0.18 | **0.28** | 0.18 |
| | **Out-of-domain** (WRITINGPROMPTS) | | | | | | | | | | | |
| **Less Toxic** | **0.18** | 0.06 | **0.25** | 0.08 | **0.16** | 0.11 | **0.16** | 0.07 | **0.16** | 0.10 | **0.15** | 0.08 |
| **More Topical** | 0.20 | 0.20 | **0.31** | 0.23 | **0.34** | 0.19 | **0.36** | 0.19 | **0.29** | 0.27 | **0.32** | 0.17 |
| **More Fluent** | **0.26** | 0.21 | **0.31** | 0.23 | **0.41** | 0.14 | **0.38** | 0.21 | **0.33** | 0.23 | **0.32** | 0.20 |

Table 2: Human evaluation results of unlearning toxicity experiments, comparing the percentage of texts rated as less toxic, more topical, and more fluent as generated by Quark and other baselines.

**Relationship to prior work.** Quantized Reward Konditioning builds upon three disjoint concepts from previous work in reinforcement learning and conditional language modeling.

(1) Inspired by PPO [91], we encourage our model to stay close to a reference model using a KL-divergence penalty. The penalty in [91] approximates KL-divergence at the sequence level through a reward penalty, $\tilde{r}(x) = r(x) - \beta \log \frac{p_\theta(x)}{p_0(x)}$, while we use a differentiable loss that exactly computes the per-step KL divergence (Eq.2); this may contribute to ease of optimization. Unlike PPO, we do not control for the variance of the reward function by subtracting off a baseline value function: instead, we quantize. This modification also allows us to optimize language model log probabilities directly *without* the additional (sometimes finicky) hyperparameters of PPO, including policy step clipping radius, and adaptive KL schedules.

(2) Inspired by the Decision Transformer [9] which frames reinforcement learning as next-token prediction, we train a model capable of conditioning on the desired reward of the trajectory, prior to observing it, i.e., our reward token appears in the input of $p_\theta(y|x, r_k)$. Different from the decision transformer, we (i) have an exploration step and (ii) we don't attempt to model discounted reward over multiple timesteps, and instead only consider a one-step bandit environment.

(3) Inspired by control codes [28] we use learned embeddings as a light-weight representation of reward. Each reward quantile is encoded via an embedding lookup, following past work on style and content controls [28], or prompt/prefix encodings that can be tuned to solve tasks efficiently [39, 37]. Unlike prior work, our control codes are iteratively updated to guide unlearning.

## 3 Experiments

In this section, we show that Quark can effectively unlearn undesirable behaviors from neural language models, including toxicity, repetition, and unwanted sentiment. Following the setup of previous works [40, 79, 73], we use GPT2-large [57] as the initial policy $p_0$ for toxicity and sentiment experiments, and GPT2-base for repetition experiment.

### 3.1 Unlearning Toxicity from Language Models

Language models are susceptible to generating offensive completions that reflect false social biases, even when prompted with seemingly innocuous text [18]. We consider unlearning toxicity from GPT-

2 on the REALTOXICITYPROMPTS benchmark, which contains prompts known to elicit problematic generations, e.g., *"I cant believe the people of..."*. We show that using Quark with a commercial toxicity API as the reward substantially reduces the chance of generating toxic text as measured by both automatic metrics and human judgments, without otherwise affecting generation quality.

**Experimental setup.** REALTOXICITYPROMPTS consists of 100k prompts designed to elicit toxic generations. We follow the experimental setup of Liu et al. [40]. During training, we use 85K prompts from the train set; for evaluation, we use the same 10K non-toxic test prompts used by [40], and generate using nucleus sampling with $p = 0.9$. Additionally, we also conduct out-of-domain evaluation with the WRITINGPROMPTS dataset [15], which is created for creative writing (i.e., story generation). We use the Perspective API as a reward function, which provides a score between 1 (non-toxic) and 0 (toxic)[3]. We use $K = 5$ quantiles.

**Baselines and evaluation metrics.** We include previously reported baselines from [40], including GPT-2 (i.e., the $p_0$ model), PPLM [12], GEDI [32], DAPT [21], and DEXPERTS [40]. Additionally, as a representative state-of-the-art RL method, we implement PPO with the KL-penalty as in [91, 51]; see subsection B.1 for details.

Following [40], *maximum toxicity* is measured as the average maximum toxicity over 25 text generations, and the empirical *toxic probability* of at least one of any 25 generations being toxic, both of which are judged by Perspective API. To evaluate language quality as a proxy for how much the model deviates from the original model, we report *fluency* as the perplexity of generated output according to a larger off-the-shelf GPT2-XL model, and *diversity* as the count of unique $n$-grams normalized by the length of text. Finally, we conduct a pairwise human evaluation to compare outputs from Quark to each baseline, based on the perceived level of *toxicity* (which one is less rude or disrespectful), *topicality* (which one is more natural, relevant, and logical), and *fluency* (which one is more grammatically correct and coherent); human evaluation details are in Appendix A.

**Results.** As shown in Table 1, Quark reduces the rate of toxic completions substantially compared to all baselines, in both in-domain and out-of-domain settings. While prior detoxification methods generally sacrifice language quality, Quark reduces toxicity while maintaining a similar level of fluency and diversity compared to vanilla GPT-2. Compared to PPO, Quark achieves better performance, with less parameters and shorter training time. Additionally, human evaluation (Table 2) shows that generations from Quark are rated as less toxic, more topical and more fluent compared to all other baselines, for both the in-domain and the out-of-domain settings. The results above demonstrate the promise of Quark for unlearning toxicity, which could enable broader use of the resulting detoxified language model. Additional qualitative results are in Appendix D.

### 3.2 Steering Away from Unwanted Sentiment of Generated Texts

Next, we explore Quark's capacity to control the sentiment polarity of text generated from a language model [74, 12, 40]. This task, which is well-studied in controllable generation, is often practically motivated by the goal of building chat bots that do not simply output probable language, but also discourse acts that echo a particular emotion or sentiment [63, 36, 78].

**Experimental setup.** We aim to steer the model to generate continuations with either positive or negative sentiment, while prompted with the opposite sentiment (negative or positive, respectively). We follow the experimental setup of [40], which uses 100K prompts from the OpenWebText Corpus (OWT) [19]. During training, we use 85K prompts from the training set. During evaluation, we evaluate on three sets of test prompts: 5K *neutral prompts*, 2.5K *positive prompts* and 2.5K *negative prompts*. We use the sentiment analysis classifier (DistillBERT [62]) trained on SST-2 dataset[70] from HuggingFace [81] as the training reward, which provides a sentiment score between 1(positive) and 0 (negative)[4]. We use $K = 5$ quantiles.

---

[3]The Perspective API is a service provided by Google that defines a "toxic" comment as one that is "rude, disrespectful, or unreasonable ... that is likely to make one leave a discussion" `https://github.com/conversationai/perspectiveapi`. Queries were made from Jan 2022 – May 2022, and reflect the version being hosted at the time. The API is itself imperfect and reflects some social biases [26, 46, 64]. See section 7 for further discussion.

[4]`https://huggingface.co/distilbert-base-uncased-finetuned-sst-2-english`

| Model | Sentiment to Unlearn: NEGATIVE | | | | | Sentiment to Unlearn: POSITIVE | | | | |
|---|---|---|---|---|---|---|---|---|---|---|
| | % Positive (↑) | | Fluency (↓) | Diversity (↑) | | % Positive (↓) | | Fluency (↓) | Diversity (↑) | |
| | negative prompt | neutral prompt | output ppl | dist-2 | dist-3 | positive prompt | neutral prompt | output ppl | dist-2 | dist-3 |
| GPT2 [57] | 0.00 | 50.02 | 11.42 | 0.85 | 0.85 | 99.08 | 50.02 | 11.42 | 0.84 | 0.84 |
| PPLM [12] | 8.72 | 52.68 | 142.1 | 0.86 | 0.85 | 89.74 | 39.05 | 181.7 | 0.87 | 0.86 |
| CTRL [29] | 18.88 | 61.81 | 43.79 | 0.83 | 0.86 | 79.05 | 37.63 | 35.94 | 0.83 | 0.86 |
| GeDi [32] | 26.80 | 86.01 | 58.41 | 0.80 | 0.79 | 39.57 | 8.73 | 84.11 | 0.84 | 0.82 |
| DEXPERTS [40] | 36.42 | 94.46 | 25.83 | 0.84 | 0.84 | 35.99 | 3.77 | 45.91 | 0.84 | 0.83 |
| DAPT [21] | 14.17 | 77.24 | 30.52 | 0.83 | 0.84 | 87.43 | 33.28 | 32.86 | 0.85 | 0.84 |
| PPO [71] | 43.13 | 94.10 | 15.16 | 0.80 | 0.84 | 32.22 | 3.65 | 15.54 | 0.81 | 0.84 |
| Quark | **46.55** | **95.00** | **14.54** | 0.80 | 0.84 | **27.50** | **2.75** | **14.72** | 0.80 | 0.84 |

Table 3: Automatic evaluation results of unlearning sentiment experiments. Baseline results (except PPO) are from [40].

| | Ours vs. GPT2 | | Ours vs. PPO | | Ours vs. CTRL | | Ours vs. GeDi | | Ours vs. DEXPERT | | Ours vs. DAPT | |
|---|---|---|---|---|---|---|---|---|---|---|---|---|
| Sentiment to Unlearn: NEGATIVE | | | | | | | | | | | | |
| More Positive | **0.58** | 0.04 | **0.16** | 0.06 | **0.46** | 0.12 | **0.38** | 0.14 | **0.32** | 0.18 | **0.48** | 0.12 |
| More Topical | **0.32** | 0.07 | **0.32** | 0.26 | **0.23** | 0.16 | **0.22** | 0.19 | **0.24** | 0.17 | **0.24** | 0.12 |
| More Fluent | **0.36** | 0.10 | **0.33** | 0.28 | **0.28** | 0.23 | 0.26 | 0.26 | **0.27** | 0.23 | **0.28** | 0.19 |
| Sentiment to Unlearn: POSITIVE | | | | | | | | | | | | |
| More Negative | **0.47** | 0.14 | **0.37** | 0.21 | **0.48** | 0.18 | **0.39** | 0.31 | **0.37** | 0.29 | **0.51** | 0.12 |
| More Topical | **0.21** | 0.18 | **0.29** | 0.18 | **0.26** | 0.20 | **0.33** | 0.17 | **0.32** | 0.16 | 0.20 | 0.20 |
| More Fluent | **0.28** | 0.24 | **0.31** | 0.20 | **0.36** | 0.22 | **0.38** | 0.21 | **0.40** | 0.23 | 0.24 | 0.24 |

Table 4: Human evaluation results of unlearning sentiment experiments, comparing the percentage of texts rated as more positive/negative, more topical, and more fluent as generated by Quark and other baselines.

**Baselines and Evaluation Metrics.**    In addition to all baselines described in §3.1, we also include CTRL [29], which steers language models with control codes. For each prompt, we generate 25 continuations at evaluation time. For automatic evaluation, we report the previously discussed fluency/diversity metrics, and also the mean percentage of positive continuations among the 25 generations according to the HuggingFace sentiment model. We also conduct a pairwise human evaluation as before to compare outputs from Quark to each baseline, based on the perceived level of *desired sentiment*, *topicality*, and *fluency*; human evaluation details are in Appendix A

**Results.**    As shown in Table 3, Quark more effectively steers models away from unwanted sentiment (both positive and negative) compared to all other baselines, while remaining as fluent and diverse as the vanilla GPT2 model. Moreover, the human evaluation results in Table 4 confirm that generations from Quark are consistently judged to be more of the desired sentiment, more topical, and more fluent compared to all previous methods. Additional qualitative results are in Appendix D.

## 3.3   Unlearning Degenerate Repetition

Neural language models often suffer from *text degeneration*, i.e., they generate repetitive, uninformative, and dull text [79, 25]. Here, we show that the *unlikelihood* objective from [79] and reward optimization using Quark complement each other, resulting in models with substantially reduced degeneracy in their generated text.

**Experimental setup.**    Our goal is to unlearn degenerate repetition in text generation. We follow the experimental setup of [79, 73]. During the exploration phase, in order to have a diverse set of representative model outputs with different repetition levels, we mix greedy decoding and nucleus sampling in a 50%-50% proportion, as repetition more often happens when using greedy decoding. We use a *diversity* metric as the reward, to encourage a larger portion of unique n-grams in generations, defined as $diversity(y) = \prod_{n=2}^{4}(1.0 - \frac{\text{rep-n}(y)}{100})$, where $\text{rep-n}(y) = 100 \times (1.0 - \frac{|\text{unique n-grams}(y)|}{|\text{total n-grams}(y)|})$. We use $K = 8$ quantiles. Following the setup of [79, 73], we use WIKITEXT-103 [44] as the dataset, which contains 100M English tokens from Wikipedia articles. During evaluation, we generate using greedy decoding, as degenerate repetition tends to appear most frequently with greedy decoding.

| Model | Language Model Quality | | | | Generation Quality | | | | Human Eval | | |
|---|---|---|---|---|---|---|---|---|---|---|---|
| | ppl ↓ | acc ↑ | rep ↓ | wrep ↓ | rep-2 ↓ | rep-3 ↓ | div ↑ | mauve ↑ | fluency ↑ | coherence ↑ | overall ↑ |
| MLE [73] | 24.23 | 39.63 | 52.82 | 29.97 | 69.21 | 65.18 | 0.04 | 0.03 | 1.89 | 2.55 | 1.96 |
| Unlikelihood [73] | 28.57 | 38.41 | 51.23 | 28.57 | 24.12 | 13.35 | 0.61 | 0.69 | 2.90 | 3.19 | 3.00 |
| SimCTG [73] | **23.82** | 40.91 | 51.66 | 28.65 | 67.36 | 63.33 | 0.05 | 0.05 | 1.93 | 2.68 | 2.08 |
| Quark | 26.22 | **41.57** | 45.64 | 25.07 | 39.89 | 30.62 | 0.35 | 0.74 | 2.75 | 3.20 | 2.77 |
| +Unlikelihood | 27.97 | 39.41 | **37.76** | **19.34** | **18.76** | **12.14** | **0.67** | **0.82** | **3.92** | **4.04** | **3.87** |

Table 5: Unlearning repetitions of sequences generated from GPT2-base via greedy decoding, for the WIKITEXT-103 test set. Baselines results are adopted from [73].

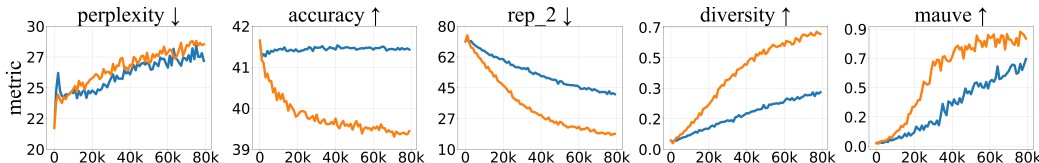

Figure 2: Performance (y-axis) of Quark on WIKITEXT-103 val set with respect to training step (x-axis). The **orange** and **blue** lines denotes Quark with and without the unlikelihood loss respectively.

**Baselines and evaluation metrics.** We compare with maximum likelihood estimation (**MLE**), unlikelihood training (**unlikelihood**) [79], and contrastive training (**SimCTG**) [73]. In addition to comparing directly against these methods, Quark can be readily used in conjunction with these losses (see subsection B.3 for details).

Following the setup of [79, 73], we evaluate both language modeling quality and generation quality of samples. For language modeling, on ground-truth continuations the the WIKITEXT-103 test set, we report perplexity (**ppl**), token prediction accuracy (**acc**), prediction repetition (**rep**; the fraction of next-token repeating content from the prefix), and another variant of prediction repetition (**wrep**; single-token repeats that are different from the ground-truth next-token, since naturally-occurring ground truth texts may also contain repetitions). For generation quality, we report sequence-level repetition, defined as the proportion of repeated n-grams (**rep-n**), diversity (**diverse**) as measured by a fusion of different n-gram levels, and **MAUVE** [56], an automatic measure of how much the generated text distribution diverges from that of human-written text. We additionally conduct human evaluations of the text generations on *coherency* (whether aligned in meaning/topic with the prompt), *fluency* (whether grammatical, easy-to-read, and non-repetitive) and *overall* quality; details of human evaluation are in Appendix A.

**Results.** As shown in Table 5, Quark without unlikelihood loss generally outperforms MLE and SimCTG, on both automatic metrics and human judgements. Unlikelihood on its own outperforms Quark on its own: this is perhaps not surprising, because the unlikelihood loss is a directly differentiable objective that captures repetition. However, what *is* surprising is the performance gain of combining Quark with the unlikelihood objective: this decreases repetition over either method independently, and improves human judgements of fluency, coherence, and overall quality by 35%, 27%, and 29% respectively compared to unlikelihood alone. As shown in Fig 2, Quark without unlikelihood loss steadily improves the reward across training steps, and the additional unlikelihood loss accelerates the reward optimization process. Additional qualitative results are in Appendix D.

## 4 Model Ablations

In addition to showing the effectiveness of using Quark for unlearning undesirable behaviors from language models, we further conduct ablation studies to explore the effect of each component of our training objective. We focus on the toxicity unlearning task for our ablation studies.

**What effect does the KL term have?** Fig 3 illustrates the effect of increasing the KL coefficient $\beta$ (our default value is $\beta = .05$), which encourages $p_\theta$ to stay closer to $p_0$. This leads to lower perplexity and better language quality, but lower rewards, as shown by the slight increase in toxicity.

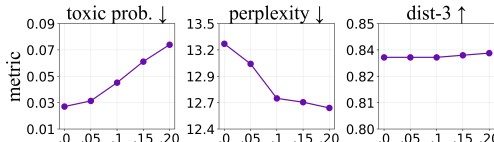

Figure 3: Performance of Quark (y-axis) on RE-ALTOXICITYPROMPTS val set, with varying KL coefficient $\beta$ (x-axis).

Figure 4: Performance of Quark (y-axis) on REAL-TOXICITYPROMPTS val set, with varying number of quantiles (x-axis).

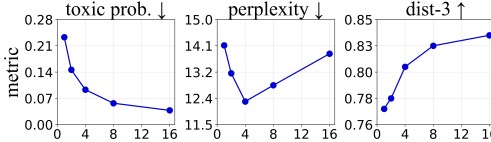

Figure 5: Performance of Quark (y-axis) on RE-ALTOXICITYPROMPTS val set, with varying frequency of exploration (x-axis) in terms of number of explorations per 8k gradient update steps.

Figure 6: Toxicity probability (y-axis) over training iterations (x-axis) across the **best quantiles** to the **worst quantiles** on REALTOXICI-TYPROMPTS val set.

| KL term | Toxicity (↓) avg. max. prob. | | Fluency (↓) output ppl | Diversity (↑) dist-2 dist-3 | |
|---|---|---|---|---|---|
| without | **0.192** | **0.031** | 13.29 | 0.79 | 0.83 |
| approx. | 0.194 | 0.038 | 13.86 | 0.80 | 0.84 |
| exact | 0.194 | 0.035 | **12.72** | 0.79 | 0.83 |

Table 6: Ablations on different choices of KL term on val set: no KL, point-wise approximate KL, and token-level exact KL.

| Explore strategy | Learn quantile | Toxicity (↓) avg. max. prob. | | Fluency (↓) output ppl | Diversity (↑) dist-2 dist-3 | |
|---|---|---|---|---|---|---|
| best-tok | all | 0.194 | 0.035 | 12.72 | 0.79 | 0.83 |
| random-tok | all | 0.286 | 0.109 | **12.40** | 0.80 | 0.84 |
| best-tok | best | **0.115** | **0.014** | 21.92 | 0.43 | 0.66 |
| $p_0$ | all | 0.291 | 0.183 | 12.53 | 0.78 | 0.80 |
| no-tok | best | 0.263 | 0.146 | 14.19 | 0.73 | 0.77 |

Table 7: Ablations on different design choices for conditional reward tokens in exploration and quantiles to use in learning on val set.

**Exact KL vs. Approximate KL.** Table 6 compares the effect of our exact token-level KL as defined in Eq.2 against an approximate point-wise KL, $\log \frac{p_0(\cdot|y_{<t},x)}{p_\theta(\cdot|y_{<t},x,r_k)}$, proposed by [71]. Compared to no KL term, the exact KL gives a controllable trade-off between language quality and reward maximization, unlike the point-wise KL, which hurts both dimensions. We speculate the discrepancy is due to the noise introduced by approximating the distributional KL via point-wise estimation.

**What effect does the number of quantiles have?** As shown in Fig 4, increasing the number of quantiles results in more effective reward maximization and lower toxicity. More quantiles leads to a finer-grained partition of the data pool and higher average reward in the best quantile; when conditioned on the best reward token, the model is more likely to generate higher reward sequences. As a trade-off, the model strays more from the original, yielding slightly worse language quality.

**Can we just train on the highest-reward quantile?** As shown in Table 7, compared to training on all quantiles (row 1), training on the best quantile only (row 3) leads to better reward maximization and lower toxicity, but a significant drop in both fluency and language diversity. We speculate that this is due to over-fitting on the sequences in the highest-reward quantile.

**Can we condition on random reward tokens in exploration?** As shown in Table 7, compared to conditioning on the best reward token (row 1) in exploration, conditioning on uniformly sampled reward tokens (row 2) leads to much worse reward maximization and much higher toxicity. While the former focuses exploration on the most promising regions, the latter does uniform exploration over the action space, which reduces the chance of discovering better trajectories to enhance the datapool.

**Are control codes useful for exploration and training?** Row 4 of Table 7 illustrates performance decreases when the initial policy $p_0$ is used for exploration instead of reward code conditioned policy $p_\theta$; Row 5 illustrates performance decreases when $p_\theta$ has no control code for both training/exploration, even when the high reward samples are added to the data pool.

**How do the rewards for generations in each partition evolve over time?**   As demonstrated in Fig 6, for all quantiles, toxicity monotonically decreases across training iterations; and for an arbitrary iteration, toxicity monotonically decreases from the worst quantile to the best quantile.

**What effect does the frequency of exploration have?**   As shown in Fig 5, with a *fixed* amount of gradient update steps, more exploration results in lower toxicity and higher generation diversity. Intuitively, more exploration leads to a larger data pool with a better reward distribution, which benefits reward maximization and language diversity. Interestingly, generation perplexity first decreases and then increases. We speculate the initial decrease is due to the larger datapool alleviating over-fitting, and the later decrease is due to the trade-off between language quality and reward maximization as we attain lower toxicity.

## 5   Related Work

**Reinforcement Learning in NLP.**   Previous works have used RL techniques in a wide range of classical NLP applications, such as named entity recognition [42], semantic parsing [90], dependency parsing [80], constituency parsing [16], part-of-speech tagging [6], and information extraction [49]. Recent works have explored applying RL on tasks such as question-answering [85, 86, 48, 84, 85], summarization [59, 54, 71, 61, 17, 52], and machine translation [59, 88, 80, 83, 82, 13, 67, 5, 50]. Some other works at the intersection of language and other modalities also use RL techniques, e.g., navigation [77, 76], multi-agent communication [35], image captioning [59, 6, 60], etc. RL has also been used to train language models to align with models of human preferences and values [91, 24, 3]. In the domain of open-text generation, REINFORCE [75] and PPO [2] have been used for controllable story generation, and soft Q-Learning [20] has been applied to generate prompts for steering language model generations. Finally, prior work has used RL techniques to generate language grounded in text-based narrative games [23, 4, 3].

**Reinforcement learning with transformers.**   Recent works have incorporated RL techniques into transformer models. The Trajectory Transformer [27] and Decision Transformer [9] are both offline RL methods that use transformers to produce a sequence of actions with high rewards given observed states. Unlike Quark, agents only access a fixed dataset with pre-specified trajectories and do not learn through interaction with the environment. Zheng et al. [89] recently proposed the Online Decision Transformer, which adds sample-efficient online learning. [72] uses PPO to incorporate human feedback for summarization.

**Unlearning undesirable behaviors from language models.**   Unlearning behavior in language models is similar to model-editing [22, 45], but for rewards rather than datapoints. Some recent works use RL for post-hoc modification of language models, e.g., unlearning toxicity [14] or non-normative generations [55]. Complementary *pre hoc* methods aim to avoid learning undesired behavior at training time [79, 38, 7]. Similarly, methods for controlling models at inference time, e.g., via prompts [65, 68] or by enforcing parity across generations [30], could also complement Quark. [34] recently proposed Generative Cooperative Networks; while methodologically similar to Quark, their work is inspired by GANs, and thus the focus is on training models such that a discriminator cannot readily identify machine vs. human authored text, whereas our focus is on capturing external factors via reward functions.

## 6   Conclusion

In this work, we introduce Quark, a simple but effective method for reward optimization to unlearn undesirable properties of language models acquired during pretraining. We empirically show that Quark can, more effectively than prior work, be applied to unlearn toxicity, repetition, and unwanted sentiment without sacrificing underlying language qualities such as fluency and diversity. Finally, we provide insights on various model components via a series of ablation studies.

Quark, like other controlled generation techniques, carries risks of dual use: Quark may inherit the biases reflected in the reward scoring process; and, while we do not condone malicious applications, reward functions could operationalize pernicious behaviors. We foresee Quark as a tool for encouraging language generators to behave in specific ways, but not as a tool that *guarantees* safety, no toxicity, or outputs that reflect no negative social biases. We discuss further in Section 7.

Future directions include:

1. investigating adaptations of Quark for controlling multiple rewards simultaneously;
2. exploring more diverse types of rewards, e.g., those related to human preferences;
3. and training Quark with fewer parameters vs. optimizing all model parameters.

## 7 Additional Ethical Considerations

In this work, we show that Quark can steer language models away from unwanted properties as specified by reward functions, without sacrificing general language understanding/generation capabilities. We foresee two primary dual use concerns for this method.

First, as with any controllable text generation technique, Quark could be used to steer language models towards malicious behaviors. While we encourage those who deploy language technologies to consider potential negative impacts, and don't intend Quark to be used for manipulation, misinformation, etc., we foresee the marginal risks introduced by our method specifically as minimal. Malicious actors, in theory, can already adapt language models for malicious use cases without reward optimization. Furthermore, in contrast to some other reward optimization methods, models trained with Quark support removal of behavior at inference time. Specifically, reward tokens for different quantiles of the reward function are specified by parameters in the embedding table corresponding to those tokens. Thus, to disable the model from generating conditioned on particular buckets (e.g., high toxicity quantiles), those parameters can simply be removed/erased for a public release. *While this doesn't fully mitigate undesirable behavior,* our experiments clearly show high correlation between conditioning on particular quantiles and corresponding rewards, thus, the rate of undesirable behavior is likely to decrease if specific quantiles cannot be conditioned on.

Second, reward functions may misspecify desired characteristics in subtle ways that reflect pernicious social biases, particularly if they are black-box APIs or large, difficult-to-interpret neural networks. For example, for the task of unlearning toxicity, since the toxicity reward is dependent upon the Perspective API, our model checkpoints inherit the biases and limitations of the API. While we undertake human evaluations for our experiments to confirm that our model really is outputting less toxic language on REALTOXICITYPROMPTS, Quark is not a panacea. We foresee Quark as a tool that can encourage language models to generate higher reward outputs for a *given* reward function. As more accurate, specific, and inclusive classifiers are built (e.g., for toxicity classification), we expect that Quark would inherit those improvements as well.

## 8 Acknowledgements

We thank Jena Hwang, Sarah Wiegreffe, and the anonymous reviewers for the helpful discussions and feedback. Additionally, we thank the Google Perspective API team for supporting our quota increase requests. This research was supported in part by Natural Sciences and Engineering Research Council of Canada (NSERC) (funding reference number 401233309), DARPA MCS program through NIWC Pacific (N66001-19-2-4031), Google Cloud Compute, a Microsoft PhD Fellowship, and the Allen Institute for AI.

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
