# OpenReview forum: "QUARK: Controllable Text Generation with Reinforced Unlearning"
_NeurIPS.cc/2022/Conference — NeurIPS 2022 Accept_

### Official Review · Reviewer_d6FS · 2022-07-09

**Rating:** 9
**Confidence:** 4
**Soundness:** 4 excellent
**Presentation:** 4 excellent
**Contribution:** 4 excellent

**Summary:**

**What is the task?**
Task of unlearning undesirable properties of language models acquired during pretraining by fine-tuning the language model on signals of what not to do.


**What has been done before?**
* In contrast to strong contemporary RL methods that stabilize training with an additional parameterized model (more learnable parameters) and specialized optimization heuristics, Quark’s training relies only on standard language modeling primitives. Experiments across three tasks demonstrate that Quark maintains pre-training abilities while unlearning undesired behaviors more stably than alternative methods.

* Quark takes inspiration from three disjoint concepts from previous work in reinforcement learning and conditional language modeling but builds upon each concept in a novel manner.

**What are the main contributions of the paper?**
* Introduced Quark, an algorithm for optimizing a reward function that quantifies an (un)wanted property, while not straying too far from the original model.

* For unlearning toxicity, negative sentiment, and repetition, our experiments show that Quark outperforms both strong baselines and state-of-the-art reinforcement learning methods like PPO, while relying only on standard language modeling primitives.

* In addition to showing the effectiveness of using Quark for unlearning undesirable behaviors from language models, author(s) conducted ablation studies to explore the effect of each component of the training objective.

**What are the key techniques used to tackle this task?** An online, off-policy reinforcement learning (RL) algorithm used to (un)learn properties from language models via three iterative stages: exploration, quantization, and learning.Quark alternates between (i) collecting samples with the current language model, (ii) sorting them into quantiles based on reward, with each quantile identified by a reward token prepended to the language model’s input, and (iii) using a standard language modeling loss on samples from each quantile conditioned on its reward token, while remaining nearby the original language model via a KL-divergence penalty.  By conditioning on a high-reward token at generation time, the model generates  text that exhibits less of the unwanted property.

**What are the main results?**
For unlearning toxicity, negative sentiment, and repetition, our experiments show that Quark outperforms both robust baselines and state-of-the-art reinforcement learning methods like PPO, while relying only on standard language modeling primitives.



**Questions:**

Is it possible to obtain significance tests for table 1 and table 3 since difference between Quark and PPO look small.

**Limitations:**

Authors have adequately addressed the limitations and potential negative societal impact of their work.

**Strengths And Weaknesses:**

Strengths

* Novel method for reward optimization to unlearn undesirable properties of language models acquired during pretraining
* SOTA automatic as well as human evaluation results on 3 different tasks
* Very nice insights on various model components via a series of ablation studies.

---

> ### Author Response · Authors · 2022-08-02
> **Response to Reviewer d6FS**
>
> Thank you for your thoughtful and constructive review as well as encouraging comments!
>
> ### Significance tests?
> Thank you for the suggestion, we will add a full significance analysis in our final draft. For example, we apply the Wilcoxon signed-rank rank test between Quark and PPO for in-domain toxicity (Table 1, left). For the metrics on which Quark achieves best values, (average max toxicity, toxicity probability, and perplexity) we find significance values against PPO of 3.7e-91, 0.039, and 2.5e-5 (respectively)–indeed, though the differences are small, they are significant. (Significance value of 0.05 or lower is generally considered statistically significant.) We will apply similar analysis across tables 1 and 3 for our final draft.
>
> Plus, despite the smaller difference between Quark and PPO on in-domain toxicity reduction, Quark has a larger gain in out-of-domain toxicity reduction (Table 1, right). Also, PPO needs more memory and runtime compared to Quark, particularly due to the reliance on the additional value network.

---

### Official Review · Reviewer_BoVo · 2022-07-10

**Rating:** 7
**Confidence:** 3
**Soundness:** 3 good
**Presentation:** 3 good
**Contribution:** 2 fair

**Summary:**

This work proposes a reinforcement learning algorithm to achieve specific goals of language model generation, by incorporating three stages: exploration, quantization and learning. In particular, the language model is finetuned with a KL-divergence penalty. In generation stage, the language model conditions on high-reward token so that the not-expected property will be less generated. Empirical results show its effectiveness on unlearning toxicity, negative sentiment, and repetition, evaluated by both automatic evaluation metrics and human evaluation.

**Questions:**

1. What is the time complexity of the algorithm? How much more time will it need compared to baselines?

2. Does QUARK has a high requirement for memory? e.g. how large is expected for the data pool to store the (input, output, reward) examples?

**Limitations:**

See weakness

**Strengths And Weaknesses:**

Strengths
1. Clear presentation, easy to follow
2. Comprehensive ablation analysis to show effects of hyperparameters and different settings

Weakness

The current framework requires specific reward design for different goals (i.e.unlearning unwanted sentiment or toxicity). In general, language generation requires many desired characteristics and the current design may not serve as an appropriate tool for this goal.

---

> ### Author Response · Authors · 2022-08-02
> **Response to Reviewer BoVo**
>
> Thank you for your thoughtful review and helpful feedback!
>
> ### Multiple rewards at once
> This is a great idea — and one that we mentioned in the conclusion session as a promising line of future work! (line 293). The flexible design of quark makes this feasible: we can use multiple control tokens corresponding to different characteristics to optimize them simultaneously. We are actively working on this now. However, we think this direction deserves its own separate paper since we focus specifically on unlearning certain unwanted behavior (e.g. toxicity) in this paper.
>
> ### Memory/runtime requirements
> Quark requires *less* runtime and memory compared to the most competitive baselines: i) compared to PPO which requires maintaining 3x copy of model weights (policy, p0, value network) we only require 2x (policy, p0); and ii) compared to decoding-based methods (PPLM, GeDi, Dexperts), quark requires only standard LM sampling for generation, whereas others require sampling and additional per-token forward and/or backward passes.
>
> Quark doesn’t have a high requirement for memory. Asymptotically, the size of the datapool is O(k*p) where k is the number of training iterations, and p is the number of exploration samples per iteration. These are hyperparameters that can be adjusted in memory limited settings. In practice, even for very large datapools, string storage of the pool is *negligible*, e.g., on the order of a few Mb.
>
> Asymptotically, the runtime complexity is also O(k*p), for each iteration we pass over our data once to get a reward, and pass over each datapoint for stochastic gradient updates. In practice, training time is dominated by the usual fine-tuning of the language model (and is readily accelerated using hardware accelerators like GPUs/TPUs).

---

### Official Review · Reviewer_5ES9 · 2022-07-11

**Rating:** 8
**Confidence:** 4
**Soundness:** 3 good
**Presentation:** 3 good
**Contribution:** 3 good

**Summary:**

Large-scale language models may generate text with offensive or toxic language, or repetition. The authors propose an algorithm, named Quantized Reward Konditioning (Quark), to unlearning these misalignments by fine-tuning the language model on signals of what not to do. Quark optimizes a reward function that quantifies an (un)wanted property and contains three steps: exploration, quantization, and learning. The experimental results on unlearning toxicity, negative sentiment, and repetition show the effectiveness of the proposed algorithm.

**Questions:**

N/A

**Strengths And Weaknesses:**

Strengths
- The authors propose a novel task to unlearn behaviors misaligned with user expectations and design a simple and effective algorithm to achieve this goal.
- The proposed algorithm can rely on standard language modeling primitives to conduct training without additional models or specialized heuristics to stabilize training.
- Extensive experiments show that Quark can be applied to unlearn toxicity, repetition, and unwanted sentiment without sacrificing underlying language qualities such as fluency and diversity.

---

> ### Author Response · Authors · 2022-08-02
> **Response to Reviewer 5ES9**
>
> Thank you for your thoughtful and encouraging comments!

---

### Official Review · Reviewer_wjcx · 2022-07-12

**Rating:** 6
**Confidence:** 3
**Soundness:** 3 good
**Presentation:** 4 excellent
**Contribution:** 3 good

**Summary:**

This paper proposes an approach to training language models to unlearn various undesirable behaviors, using a reward function. The authors propose to do this by iterating the final 3 steps of the following procedure: sample from the model and obtain corresponding reward values, sort the resulting generations into k-quantiles, fine-tune the model on these generations conditioned on the corresponding k-quantile, but with a KL-penalty that prevents the model from deviating too far from the initial model, and then re-sample generations conditioned on the highest k-quantile. The authors evaluate their approach in its ability to prevent generation of toxic text, to flip sentiment, and to avoid repetition. They find their approach outperforms baselines in terms of automatic and human evaluation.

**Questions:**

For the ablation experiments in Table 7: does the experiment in the last row, where the model is trained just on the "best" quantile, ensure that the model still sees as much data as the models in the previous two rows which are trained on "all" the quantiles?

**Limitations:**

Yes, the authors address limitations and potential negative societal impacts adequately.

**Strengths And Weaknesses:**

Strengths:
- The paper is clearly written.
- The proposed approach is simple and intuitive.
- The authors do a thorough evaluation, and the results are impressive.
- The authors include very interesting ablation experiments.

Weaknesses:
- Although the authors consider many ablations, since Quark does seem quite similar to PPO-like approaches, I think we could have benefited from seeing the results of some other baselines, including:
    - Fine-tuning on the resampled outputs (from the top quantile) but without conditioning them on the quantile control token. This tests whether using control tokens is important.
    - Doing “exploration” with the original model $p_0$ rather than $p_\theta$, which simply amounts to fine-tuning (with control tokens, say) on a larger corpus that has been labeled by the reward function. This seems like the simplest fair baseline, and checks whether it is even important to keep the samples up-to-date with the current model (which is in any case being constrained to stay close to $p_0$).
- Similar to the previous point, it’s not clear to me that the baselines from prior work are especially fair. For instance, DAPT is somewhat similar to the proposed second baseline above, except it is fine-tuned on a different corpus. Presumably this baseline would be more competitive if it were fine-tuned using results of the toxicity classifier being used to evaluate, as Quark is.
- If the paper is interested in positioning itself as being about an RL technique or reward maximization, I would expect experiments to report the actual reward achieved by Quark or any of the baselines.

Update after author response: thanks for your responses and for the new results. I'm increasing my score.

---

> ### Author Response · Authors · 2022-08-02
> **Response to Reviewer wjcx (part 1)**
>
> Thank you for your detailed review, insightful and constructive comments!
>
>
> ### Additional suggested baselines
>
> Based on your feedback, we ran some additional experiments. Specifically: we ran the additional suggested ablations — Quark without the control code, and using the initial policy for exploration instead of the updated policy.
>
> | Model | Avg. max. toxicity ↓ | Toxicity prob. ↓ | perplexity ↓ | Dist-2 ↑ | Dist-3 ↑ |
> |-------------------|----------------------|------------------|--------------|----------|----------|
> | quark | **0.196** | **0.035** | **12.47** | **0.80** | **0.84** |
> | w/o control code | 0.263 | 0.146 | 14.19 | 0.73 | 0.77 |
> |exploration with p0| 0.291 | 0.183 | 12.53 | 0.78 | 0.80 |
>
> As shown above, fine-tuning on the resampled outputs from the top quantile without conditioning them on the control token (row 2) leads to much worse toxicity reduction and worse language diversity;
>
> Doing “exploration” with the original model p0 rather than pθ (row 3) leads to dramatic performance drops in toxicity reduction. Although pθ is constrained to stay close to p0, pθ is optimized to be a better policy and more likely to obtain trajectories with higher reward in exploration. As we shown in Figure 6, when we use updated pθ for exploration, the expected toxicity of each quantile in the data pool monotonically decreases across training iterations; whereas when we use p0 for exploration, the expected toxicity in the data pool should stay the same.
>
>
> ### On the fairness of our baselines
>
> We consider 6 strong baselines in our paper, spanning two main categories: decoding-based (PPLM, GeDi, Dexperts) and learning-based (CTRL, DAPT, PPO).
>
> On the suggestion of using the same toxicity classifier for the DAPT baseline as is used for Quark–in fact, DAPT already uses **the same toxicity classifier** (perspective api) as Quark, which provides the signal for filtering non-toxic corpus for fine tuning. Thus the source of supervision is comparable between these methods. Similarly, for the sentiment controlling task, DAPT uses the same sentiment classifier as Quark. For all learning based baselines, the reward signal uses the same toxicity or sentiment classifier.
>
> While we include decoding-based methods as baselines because they are previous SOTA on the benchmarks we tested on, these are nontrivial or sometimes impossible to make completely comparable with Quark. We report performance as they were proposed in original papers because they require additional assumptions that we don't have (e.g. PPLM needs the classifier to be differentiable; GeDi, Dexperts need domain-specific (e.g. toxic) language models), which makes it impossible to adapt our simple blackbox classifier as their supervision.
>
> ### Report the rewards attained
>
> Our work is focussed specifically on unlearning unwanted behavior, rather than general reward maximization. As such, we report values that are related to reward achieved, but more focussed on practical attributes of systems. For instance, in the toxicity setting we report average per-example maximum toxicity which better captures risk than simply reporting (inverse) reward, i.e. mean toxicity. We tend to follow past work in the scores which we report, which are quite related to reward but with a more practical interpretation.
>
> We are happy to include reward attained by each method in our final draft. We expect trends to be quite similar to the values we already report–for instance, comparing between average maximum toxicity (reported in table 1) and mean toxicity (negative reward), we see the same trends, with Quark achieving the lowest value (toxicity=0.067), followed by PPO (toxicity=0.074), with GPT-2 generating the highest toxicity (toxicity=0.16). This ordering matches already reported scores. We will include reward across experimental settings and methods in our final draft.

---

> > ### Author Response · Authors · 2022-08-02
> > **Response to Reviewer wjcx (part 2)**
> >
> > ### Whether the same amount of data is used for "best quantile" ablations
> >
> > Reviewer wjcx asks whether the “best quantile” method in table 7 adjusts data generation to assure the model is trained on the same amount of data as other versions in this table. In fact, this would require a significantly larger amount of *generation and supervision* than the other rows in this table (N times as much where N is the number of quantiles), as this method discards most generations and supervision it has access to. To keep the ablations fair in terms of **resources** each version is allowed, we hold the amount of generations and supervision constant i.e. asking “what can each method do with a static budget of supervision and generation?” Thus the “best” trains on 1/N the data as it discards much of its supervision.
> >
> > However, we agree that the “best quantile” version with the same amount of *training data* is also an informative ablation, and so we add this ablation below (row 3). The results are seemingly quite similar to “best quantile” using only 1/N of the data (row 2), suggesting that the difference in data size is not a significant factor in full Quark outperforming the “best quantile” method.
> >
> > | Quantile for learning | Avg. max. toxicity ↓ | Toxicity prob. ↓ | perplexity ↓ | Dist-2 ↑ | Dist-3 ↑ |
> > |-------------------|----------------------|------------------|--------------|----------|----------|
> > | all  | 0.196 | 0.035 | **12.47** | **0.80** | **0.84** |
> > | best (same amount of resource) | **0.115** | **0.014** | 21.92 | 0.43 | 0.66 |
> > | best (same amount of data)| 0.165 | 0.028 | 22.14 | 0.64 | 0.76 |

---

### Author Response · Authors · 2022-08-02
**Summary Response**

We thank reviewers for their thoughtful reviews and constructive comments. We appreciate the encouraging remarks, e.g., that Quark’s performance is “impressive” (wjcx), that Quark achieves “SOTA automatic as well as human evaluation” (d6FS), and that our ablations/experiments are “comprehensive”/”extensive” (BoVo/5ES9).

In addition, we provide additional baselines (wjcx), clarify the fair comparison with previous works (wjcx), provide runtime and analysis (BoVo), conduct significance tests (d6FS), and address other concerns/questions raised by reviews. Please let us know if there are additional concerns or questions.

---

### Meta-Review · Area_Chair_dWko · 2022-08-21

**Recommendation:** Accept
**Confidence:** Certain

**Metareview:**

This paper proposes Quantized Reward Konditioning (Quark), an algorithm for (un)learning language model misalignments. This is an important research direction given the importance of developing better-aligned large language models, and the paper does a good job at presenting why it matters and how it is related to prior work. The reviewers think that the paper is well written and clear, and that the approach is novel, sound and interesting. After the rebuttal, all reviewers vote to accept.

**Award:**

No

---

### Decision · Program_Chairs · 2022-09-14

Accept